# Effects of a Multifactorial Program with Case Management for Falls Prevention on Functional Outcomes in Community-Dwelling Older People: A Randomized Clinical Study

**DOI:** 10.3390/healthcare12151541

**Published:** 2024-08-03

**Authors:** Areta Dames Cachapuz Novaes, Juliana Hotta Ansai, Silsam Napolitano Alberto, Maria Joana Duarte Caetano, Paulo Giusti Rossi, Mariana Luiz de Melo, Karina Gramani-Say

**Affiliations:** 1Graduate Program in Gerontology, Federal University of São Carlos, São Carlos 13565-905, SP, Brazil; jhansai@ufscar.br (J.H.A.); silsam@estudante.ufscar.br (S.N.A.); marianamelo@estudante.ufscar.br (M.L.d.M.); gramanisay@ufscar.br (K.G.-S.); 2Paulínia City Hall, Paulínia 13140-000, SP, Brazil; joanadcaetano@hotmail.com; 3Research Laboratory of Older Adults’ Health (LaPeSI), Department of Physical Therapy (DFisio), Federal University of São Carlos (UFSCar), São Carlos 13565-905, SP, Brazil; paulo.giusti.rossi@gmail.com

**Keywords:** accidental falls, aged, risk management, physical exercise

## Abstract

Falls are among the top 10 causes of years lived with disability in people aged 75 and over. Preventive programs like case management (CM) are crucial. Objectives: To evaluate the effects of a multifactorial fall prevention program based on CM on physical performance, the presence of pain, and the risk of falls and fractures in older people who have suffered falls. Methods: This randomized, single-blind clinical trial with parallel groups, Intervention Group (IG) and Control Group (CG), was composed of 55 older people with a history of falling, living in the community. All participants underwent an initial assessment via video call (containing anamnesis, timed up-and-go test, falls risk score, short physical performance battery, and clinical frax). The IG underwent CM, the physical exercise protocol, and the cognitive stimulation protocol. The CG was monitored through telephone calls and received general health and fall guidance. Results: No significant results were found in the physical capacity, the presence of pain, the risk of falls, or the fractures between the Intervention and Control Groups and between assessments. Conclusion: This program was not effective in improving functional performance, but it was important for characterizing pain and the probability of fracture in the next 10 years in this population.

## 1. Introduction

Accidental falls are one of the most common health problems related to aging. The fall rate can vary between 28% and 35% in people over 65 years old and around 40% among people over 80 [1]. These accidents can negatively impact the quality of life, socialization, family dynamics, health costs, fear of falling, and risk of death [2,3]. Falls are associated with multifactorial circumstances that can act in isolation or together, including intrinsic (such as sarcopenia, functional loss, low visual and auditory acuity, cognitive decline, postural instability, balance and gait impairments, polypharmacy, musculoskeletal pain, and chronic diseases) and extrinsic risk factors (such as environmental conditions, uneven and slippery floors, insufficient lighting, arrangement of furniture, steps, public roads and sidewalks with holes and irregularities, and lack of handrails and signage) [4,5,6,7,8,9].

Pain is also a significant risk factor for falling. One in two older people with musculoskeletal pain falls each year [10]. This occurs due to neuromuscular control changes and compensatory movements, which impact balance and gait, promote muscle weakness and joint injury, and decrease executive function performance [10]. Furthermore, pain is little investigated among older people. At least 53% of ambulatory patients with moderate pain and 30% with severe pain do not receive any diagnosis [11].

Therefore, it is essential to encourage fall prevention programs with available, efficient, and low-cost strategies, such as health education, home environment, physical activity practice, medication review, vitamin D supplementation, and health reviews. Multifactorial interventions are important to reduce risk factors for falls and their consequences, to improve physical and cognitive aspects, and to promote aging in place, especially among taller and more frail older people [9,12,13,14,15,16,17].

Multifactorial interventions can prevent and reduce fall risk in community-dwelling older people [15]. When associated with the prescription and performance of specific physical exercises, it can reduce the fall rate [15]. These interventions work with a multidimensional individual assessment to verify modifiable risk factors for falls and with specific referrals and strategies to reduce the identified risk factors for each person [18]. Physical exercise, when associated with multifactorial interventions, can have better adherence and promote long-term behavior change in older people [19].

Case management is a new multifactorial intervention approach with the possibility of active discussions between participants, caregivers, family members, and healthcare professionals about the risk factors identified in a multidimensional assessment. Also, case management protocols create and implement individualized and personalized plans to reduce risk factors for falls and their consequences [20,21]. Although most fall prevention programs include an assessment of fall risk factors and physical exercise, they do not include case management as a core element of the intervention program [21]. Therefore, this study aims to evaluate the effects of a multifactorial fall prevention program, based on case management, on physical performance, the presence of pain, and the risk of falls and fractures in older people with a history of falls. Our hypothesis was that the participants in the case management intervention would present better results in terms of physical performance, the presence of pain, and risk of falling compared to the group that received only monthly monitoring.

## 2. Materials and Methods

### 2.1. Trial Design and Setting

This is a block randomized design (allocation ratio 1:1), parallel-group, single-blind (assessors) controlled trial. The trial was approved by the Research Ethics Committee of the Federal University of São Carlos (34350620.7.0000.5504), and it was registered in the Brazilian Registry of Clinical Trials (RBR-3t85fd). A pilot study was conducted to identify possible barriers and facilitators to adhesion to the program and to standardize all procedures.

### 2.2. Study Population

Volunteers interested in participating in the study initially went through an eligibility stage, a quick screening to verify whether the volunteer’s profile was aligned with the main inclusion criteria of the study. Therefore, Brazilian adults living in the community, aged 60 or over, non-institutionalized, and with a history of falls in the last year went through this stage. Only after this screening were the inclusion and exclusion criteria duly verified in detail.

The inclusion criteria were a history of at least twice in the last year, the self-reported ability to walk with or without walking devices, and an older people or a family member with a prior, self-reported relationship with the use of remote devices. The exclusion criteria were the self-reported presence of severe and uncorrected visual or auditory disorders that affected communication during the evaluation and intervention. Furthermore, when asked, those who had received any prior self-reported medical diagnosis of active inflammatory and neurological diseases that severely interfered with balance performance, including advanced Parkinson’s disease (modified Hoehn and Yahr Scale stage 5) and were not on regular antiparkinsonian medication, in addition to prior medical diagnoses of multiple sclerosis, Huntington’s disease, dementia, uncontrolled vestibulopathy, epilepsy, traumatic brain injury, and severe motor sequelae of stroke, were also excluded.

### 2.3. Power and Sample Size

The sample size was calculated using the G*Power 3.1 software, based on the main variable (timed up-and-go test), the type of study design (two-way ANOVA test), the type I error at 5%, the statistical power of 80% [2,9,13,15], the moderate effect size (0.20), and the number of groups (2). The total sample size was 42 people; however, taking into account a possible dropout rate of 20%, we recruited 62 community-dwelling older adults.

### 2.4. Location and Disclosure

The entire research was conducted remotely at the volunteer’s home via video calls on Google Meet or telephone.

Volunteers were recruited through referrals from health care networks, dissemination on social networks, radio programs throughout Brazil, interviews on television programs, distribution of flyers and posters, promotional emails to higher education institutions, and health, educational, and social secretariats in all states of Brazil.

### 2.5. Randomization and Blinding

A researcher not involved in the recruitment, assessment, or intervention generated a sequence list using the random allocation software (1.0.0). According to the randomization sequence, each volunteer corresponded to an opaque and sealed envelope, numbered in order, with a card indicating which group the volunteer would be inserted into. After the initial evaluation, the volunteers were allocated (1:1) to 1 of the 2 groups: Intervention Group (IG) and Control Group (CG).

### 2.6. Assessments

Before the assessments, the researchers who administered the tests were trained by a senior researcher with years of experience in administering tests. Also, pilot tests were conducted to minimize any measurement errors. Support materials were sent to the volunteers to facilitate the use of the remote devices, provide materials, and create the space needed. A previous video call was performed to better explain the use. The assessment was carried out individually through a video call on Google Meet. Caregivers or family members helped the researcher when the older person had difficulty using a computer or cell phone during video calls. The evaluators also requested external help to position the device’s camera so that the researcher could see the volunteer’s entire body during the tests. All assessments were recorded according to prior authorization. The recordings and data were stored in Google Drive (a virtual file storage service), and only the researchers who carried out the assessment and the coordinators had access to verify information when necessary. The video security guarantee is offered by Google, and access to these videos is only released through 2-step authentication, which makes them difficult for hackers to access.

Volunteers were previously instructed to wear comfortable clothing, preferably closed-toed shoes, not perform vigorous physical exercises the day before the assessment, and to use auditory or visual aids when necessary. The assessments were carried out in 4 moments (initial, reassessment after 16 weeks of intervention, short follow-up after 6 weeks of reassessment, and long follow-up after 12 months of the initial assessment). The assessments were conducted remotely during the day or afternoon at a previously scheduled time according to the participant’s availability. If possible, the period was standardized between assessments. The variables evaluated in this study were fall risk, the risk of osteoporotic fractures over the next 10 years, physical performance, and the presence of pain.

The fall risk score was applied to assess fall risk. This scale presents 5 criteria: (1) the presence of previous falls; (2) medications used that may increase fall risk; (3) the presence of a sensory deficit; (4) the mental state through the mini-mental state examination; and (5) gait. The scale ranges from zero to 11 points, and scores greater than or equal to 3 points suggest that the person is at high fall risk [22,23].

The risk of osteoporotic fractures in the next 10 years was assessed by the clinical FRAX^®^ tool. For this calculation, it is necessary to use body mass index and age values, whether the participant presents previous fractures, whether he/she is a smoker, and the use of glucocorticoid-type medications. A FRAX^®^ value lower than 5 means a low fracture risk. Values between 5 and 7.5 mean intermediate risk, and values greater than 7.5 are considered high risk of osteoporotic fractures [24,25].

Physical performance was assessed by the short physical performance battery (SPPB), the timed up-and-go test (TUG), and TUG with dual task (TUG DT). The SPPB consists of three tests that evaluate static balance (0–4), gait speed (0–4) and lower limb muscle strength (0–4). The final SPPB score is given by the sum of the 3 scores, which can vary between 0 and 12 points: 0 to 3 points indicate disability or poor ability; 4 to 6 points indicate low capacity; 7 to 9 points indicate moderate capacity; and 10 to 12 points indicate good capacity [26]. The TUG requires that the volunteer stand up from a chair, walk 3 m in a straight line at usual speed, turn around, walk back the same 3 m, and sit down again [27]. A time of 12.47 s identifies Brazilian older people at fall risk [28]. The TUG DT included the TUG associated with saying the names of animals (verbal fluency). A time greater than 15 s in the TUG DT indicates greater fall risk in community-dwelling older people [29,30].

The presence of pain was assessed by the Brief Pain Inventory, which includes questions about the volunteer’s pain during the day, the pain points indicated by a body diagram, the pain intensity, the pain treatment or relief medication, and the interference of pain with general activity, mood, ability to walk, work, relationships with other people, sleep, and ability to enjoy life [31].

### 2.7. Intervention

After randomization, the IG participants received an intervention with individual case management to modify the identified risk factors for falls. The intervention was carried out by two previously trained researchers weekly through video calls via WhatsApp or phone calls. In the first weeks, the case manager, the participant, and their family members established a trusting relationship and discussed the main points identified in the multidimensional assessment and the older adult’s goals and preferences. Then, the case managers, with the help of the research team, built a personalized implementation plan to be carried out over the remaining weeks. This plan was fully personalized, taking into account all the risk factors identified in the multidimensional assessment. Each week, a different factor was addressed, prioritizing those previously identified. In the last week, a review of all topics worked on was carried out [32].

In addition, IG participants were invited to participate in a home-based multicomponent physical exercise program delivered through recorded video guidelines twice a week during the intervention period. The exercise program was developed based on a previous protocol [33] and in line with the main recommendations [34,35]. It was adapted to a remote format and to community-dwelling older adults with a history of falls in the primary health care setting. The physical exercises lasted around 40 min and occurred twice a week. The protocol consisted of lower limb strengthening, postural balance, coordination, dual-tasking, and gait physical exercises. Progression occurred individually through biweekly calls by the physiotherapy team. The volunteers were stimulated and trained to perform more advanced physical exercises. More information about the case management protocol is described by Alberto et al. [32].

The CG received only monthly calls during the 16 weeks to monitor falls and general health statuses.

### 2.8. Data Analysis

A significance level of 0.05 and the SPSS software (22.0) were used to perform the statistical tests by intention-to-treat. The Kolmogorov‒Smirnov normality test was applied to all continuous variables to verify data distribution. The chi-square test of association was used for categorical variables to compare groups and assessments. The independent *t*-test was used to compare groups for continuous variables at baseline. A two-way ANOVA was used to test the interaction between groups and assessments. In the case of interaction, analyses of simple main effects were performed, with adjustments for multiple comparisons (Bonferroni).

## 3. Results

The final sample consisted of 55 participants, with 28 volunteers allocated to the IG and 27 people to the CG (Figure 1). Eighty-six percent of the volunteers were from the southeastern region of Brazil, 7% from the northeastern region, 5% from the southern region, and 2% from the central western region, with no participants from the northern region.

We did not find significant differences in sociodemographic characteristics between groups at baseline (Table 1). The sample was composed of mostly women, with a mean age of 72–73 years, a mean level of education of 10 years, two diseases (mean), and three to four medications in use (mean).

### 3.1. Risk of Falls and Fractures

At baseline, the mean number of previous falls in the last 12 months was 2.66 in the IG and 3.40 in the CG, which indicates a population at high risk for falls. The results of the SPPB, TUG, FRAX, and BPI tests for each group at each assessment time are presented in Table 2. The FRAX^®^ presented a mean score of seven points in the IG and eight points in the CG at baseline, which means a high risk of fractures. No significant differences were found between groups and assessments (Table 2).

### 3.2. Physical Performance

At baseline, the mean SPPB score was five points for both groups. Also, the IG and CG participants completed the TUG test at baseline in 17 s (mean) and 20 s (mean), respectively, indicating that both groups are at high risk of falling. In the TUG DT, the average times to complete the test at baseline were 24 s and 27 s for participants in the IG and CG, respectively, indicating dual task impairments.

There were no significant differences between groups and assessments in any physical performance variable. After 16 weeks of intervention, in the short and long follow-up periods, the physical performance scores remained similar in both groups. Despite this, the CG increased the time to perform TUG DT (mean of 33 s) after the long follow-up period.

### 3.3. Presence of Pain

At baseline, 19% of the volunteers in the IG and 29% in the CG reported pain (*p* = 0.811). The most cited part of the body with pain in both groups was the lumbar spine. The mean number of pain sites was five for both groups. Pain intensity had an average score of two points for the IG and three points for the CG, using a scale that ranged from zero (no pain) to ten (maximum pain) points. There were no significant differences between groups or assessments in the presence of pain.

## 4. Discussion

This randomized controlled trial showed that, compared to CG, 16 weeks of a multifactorial fall prevention program based on case management did not improve physical performance and did not reduce the presence of pain or the risk of falls and fractures in community-dwelling older people with a history of falls. Our findings did not support our initial hypothesis, in which it was expected that participants who received the intervention based on case management would perform better.

In contrast to our findings, other fall prevention programs for community-dwelling older people presented positive results [36]. A systematic review conducted by Sherrington et al. [36] found that physical exercise that mainly involved balance and functional training reduced falls compared with an inactive control group. In the present study, the physical exercise program contained in the case management intervention was provided through recorded videos, and participants had to perform physical exercises at home. Center-based exercise programs with the presence of a health professional may be more effective than home-based exercise programs.

In a recent study, a remote home-based fall prevention program presented positive results in physical function, psychological factors, and balance among community-dwelling older adults who were enrolled at a welfare senior center [37]. In the present study, participants had a history of at least two falls in the last year, which characterizes them as a more fragile sample of older people. Furthermore, 11 of the 32 IG participants (34%) did not participate fully in the physical exercise program during the intervention period, due to a lack of interest and the absence of medical approval, as requested in the safety protocol. For these cases, light physical exercises associated with fall prevention and personal interest were recommended. Despite this, it is important to note that the main intervention in this study was the case management program.

Another factor that may have also contributed to the lack of improvement in the physical performance of the IG participants was the delay in starting the exercise program, which prevented participants from completing all training sessions. Among the 21 participants who agreed to participate in the physical exercise program, 13 completed at least 12 weeks of training. Participants completed an average of 23 sessions and six biweekly calls for progression of the physical exercise protocol. This was mainly due to the study design. During the first week of the intervention, participants were invited to take part in a physical exercise program as a case management approach to prevent future falls. Participants who agreed to take part in the exercise program were required to provide a medical certificate before starting training. The delay in scheduling appointments and exams in primary health care was an issue, preventing many participants from completing the proposed training period.

Although no significant differences were found in pain measures, the assessment allowed us to identify the lumbar spine as the part of the body with the most pain cited by the participants. Moreover, the number of pain sites decreased over time in both groups. This information may be important to characterize pain in this population and to plan new strategies for managing pain in older people who have suffered falls. Most fall prevention programs focus on assessments and interventions related to foot pain, as the feet play an important role in body balance [38]. In the present study, pain in the lumbar spine was identified as a possible risk factor for falling, and, thus, future studies should also target this variable.

Tep et al. [39] verified the effectiveness of the Otago home program on fall prevention compared to Frenkel physical exercises. Both groups presented positive results in TUG performance; however, the Otago program had positive results in risk and fear of falling. Additionally, Wu et al. [40] found improvements in balance, muscle strength, and mobility after specific fall prevention programs, including remote protocols. This study reinforces the correct choice of using specific physical exercises to complement the remote case management intervention in older people with a history of falls.

In addition, there is evidence that a home-based fall prevention Otago exercise program can reduce pain among community-dwelling older adults [41]. This study used a 1-year exercise intervention in which exercises were individually adjusted by a physical therapist during six home visits. Therefore, the presence of a health professional to determine the structure and progression of the individually tailored exercises may be crucial to reducing pain and improving performance.

The results of the present study also suggest a high risk of fractures among the recruited volunteers. The FRAX^®^ algorithm quantifies a patient’s 10-year probability of a hip or major osteoporotic fracture by using body mass index data, age values, previous fractures, whether he/she is a smoker, and the use of glucocorticoid-type medications. Such variables are not modifiable by a short-term intervention, which can explain the lack of differences between pre- and post-intervention. It is important to note that this tool does not take balance measurements into account when calculating fracture risk [24].

The fall risk score assessment is an important tool in the educational process for older people at risk of falling. In the present study, assessment results were presented and discussed with participants as a strategy for personalized execution plans. As a result, it was possible to hold conversations related to the concept of falling, the correct use of medications, reflections on the accidents that occurred, and the risk factors involved. Therefore, providing guidance on the risk factors involved in fall accidents can be interpreted as an improvement in care and, consequently, a strategy for preventing falls and fractures.

Future studies aiming at the development of fall prevention programs should consider the participant’s access to health services, familiarity with and access to technology, and the inclusion of assessment and management of pain with a larger sample size. Also, it is important to validate the assessments applied remotely to older people with a history of falling. Also, a better understanding of the effectiveness of case management interventions to prevent and reduce falls is needed, with adjustments to guarantee their implementation in clinical practice.

### Strengths and Limitations

A strength of our study was the remote application of tests and case management interventions during the COVID-19 pandemic. For this reason, all assessments and interventions were adapted to be carried out via video and telephone calls; thus, we acknowledged that in some way this may have interfered with the results. Some assessments were carried out at different times according to each participant’s availability. Also, an important inclusion criteria was that participants should be able to communicate through video calls. This may have limited the possibility of having a more diverse sample of older people. Finally, our sample comprised self-selected volunteers and relatively educated older people (average of 10 years of schooling), so the findings cannot be generalized to less educated older people.

## 5. Conclusions

The multifactorial fall prevention protocol based on case management for community-dwelling older people with a history of recurrent falls was not effective in improving physical performance, presence of pain and risk of falls and fractures. Future studies aimed at developing fall prevention programs should consider complementary strategies for older people at high risk of falling.

## Figures and Tables

**Figure 1 healthcare-12-01541-f001:**
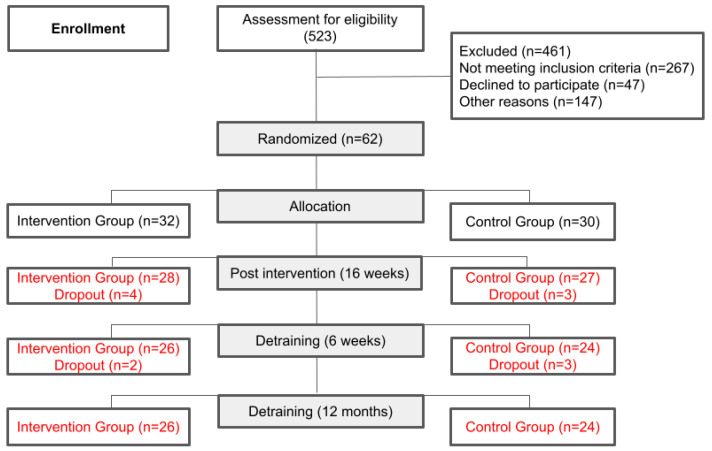
Flowchart of volunteers through the study (Consort 2010). n = number of individuals.

**Table 1 healthcare-12-01541-t001:** Characteristics of the sample at baseline (n = 55).

Variables, M ± SD or n (%)	Intervention Group (n = 28)	Control Group (n = 27)	*p* Value
Women	24 (85.71)	24 (88.88)	0.756
Age (years old)	72.06 ± 8.24	73.33 ± 9.13	0.567
BMI (kg/m^2^)	27.44 ± 5.55	27.42 ± 4.63	0.645
Years of schooling	9.78 ± 5.72	10.37 ± 5.53	0.711
Income (brazilian MW)			
Up to 1 MW	5 (17.85)	7 (25.92)	0.946
1–2 MW	6 (21.42)	6 (22.22)	
2–3 MW	3 (10.71)	6 (22.22)	
3–5 MW	5 (17.85)	3 (11.11)	
5–10 MW	6 (21.42)	3 (11.11)	
10–20 MW	2 (7.14)	2 (7.40)	
Over 20 MW	1 (3.58)	0 (0)	
Number of falls	2.66 ± 0.83	3.40 ± 2.27	0.218
Number of diseases	2.16 ± 1.25	2.27 ± 1.41	0.168
Medication in use	3.28 ± 2.78	4.57 ± 3.88	0.431
Marital status			0.423
Married	8 (28.57)	11 (40.74)	
Single	3 (10.71)	0 (0)	
Widower	15 (53.57)	11 (40.74)	
Separated/divorced	2 (7.14)	5 (18.51)	
Race			0.674
White	22 (78.57)	22 (81.48)	
Brown	5 (17.85)	3 (11.11)	
Black	1 (3.57)	2 (7.40)	

M ± SD = mean ± standard deviation; n (%) = number of individuals (percentage); BMI (kg/m^2^) = body mass index (kilogram/meter squared); MW = minimum wage.

**Table 2 healthcare-12-01541-t002:** Intergroup and intragroup analysis of risk of falls and fractures, physical performance and presence of pain (n = 55).

Variables	Intervention Group (n = 28)				Control Group (n = 27)				*p* Value		
	Baseline	16 Weeks	Short Follow-Up	Long Follow-Up	Baseline	16 Weeks	Short Follow-Up	Long Follow-Up	*p* Value Group * Time	*p* Value Group	*p* Value Time
SPPB											
Balance score	3.13 ± 1.31	2.84 ± 1.44	3.13 ± 1.16	3.16 ± 1.28	3.07 ± 1.28	3.03 ± 1.35	2.90 ± 1.35	2.80 ± 1.40	0.125	0.711	0.598
Chair stand score	1.16 ± 0.95	1.28 ± 1.05	1.25 ± 0.98	1.25 ± 0.84	1.10 ± 0.66	1.07 ± 0.82	1.20 ± 1.00	1.03 ± 0.72	0.443	0.829	0.890
Gait speed score	1.47 ± 1.13	1.66 ± 1.23	1.72 ± 1.22	1.81 ± 0.96	1.50 ± 0.82	1.73 ± 1.01	1.93 ± 0.78	1.60 ± 0.97	0.661	0.308	0.102
Total score	5.75 ± 2.48	5.09 ± 3.40	4.97 ± 3.26	5.09 ± 3.41	5.67 ± 2.07	5.13 ± 3.07	4.97 ± 3.20	4.17 ± 3.28	0.923	0.863	0.020 *
TUG (seconds)	17.28 ± 9.89	18.12 ± 11.15	18.09 ± 12.41	18.19 ± 10.36	20.66 ± 15.29	17.72 ± 10.91	20.32 ± 17.92	22.68 ± 22.39	0.308	0.546	0.478
TUG Dual task (seconds)	24.19 ± 17.30	24.91 ± 19.06	24.90 ± 19.28	25.04 ± 16.11	27.45 ± 24.55	26.24 ± 20.29	29.76 ± 33.09	33.66 ± 39.52	0.559	0.495	0.753
Clinical FRAX score	7.83 ± 4.09	7.67 ± 3.56	7.17 ± 3.96	NR	8.63 ± 6.65	7.93 ± 5.93	8.19 ± 6.08	NR	0.393	0.594	0.130
BPI											
Number of pain sites	5.03 ± 4.18	4.34 ± 4.42	4.81 ± 5.11	NR	5.77 ± 4.71	3.67 ± 4.48	4.20 ± 5.35	NR	0.057	0.740	0.001 *
Pain severity score	2.03 ± 2.96	2.25 ± 3.36	2.28 ± 3.04	NR	3.23 ± 3.92	3.03 ± 3.80	2.53 ± 3.48	NR	0.378	0.342	0.738
Fall risk score	2.09 ± 1.02	2.06 ± 1.13	1.91 ± 1.17	1.88 ± 1.26	2.30 ± 1.44	2.33 ± 1.18	2.30 ± 1.17	2.07 ± 1.25	0.846	0.324	0.223

* Significance level = *p* ≤ 0.05; ± = standard deviation; SPPB = short physical performance battery; TUG = timed up-and-go test; BPI = brief pain inventory; short follow-up = 6 weeks after reassessment; long follow-up = 12 months after randomization.

## Data Availability

The data presented in this study are available on request from the corresponding author due to privacy and ethical reasons.

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
