# Peer review of "Effects of a Multifactorial Program with Case Management for Falls Prevention on Functional Outcomes in Community-Dwelling Older People: A Randomized Clinical Study"

_healthcare, 2024, doi:10.3390/healthcare12151541_

Round 1

Reviewer 1 Report (Previous Reviewer 1)

Comments and Suggestions for Authors

I accept in present form

Author Response

Dear Reviewer 1,

We would like to thank for allowing us to resubmit our manuscript with all corrections. All the criticisms raised have been carefully addressed, as requested, including in the Results section. In the revised manuscript, the changes have been highlighted in red. We believe that the present version of the manuscript has been improved. We are fully available to answer other issues. We hope that now it is suitable for publication in the Healthcare. 

Sincerely,

Authors.

Reviewer 2 Report (Previous Reviewer 2)

Comments and Suggestions for Authors

All comments addressed by authors.

Author Response

Dear Reviewer 2,

Thank you for your comment. We hope that the article can contribute positively to the journal. We would like to thank for allowing us to resubmit our manuscript with all corrections. All the criticisms raised have been carefully addressed, as requested, including in the Results section. In the revised manuscript, the changes have been highlighted in red. We believe that the present version of the manuscript has been improved. We are fully available to answer other issues. We hope that now it is suitable for publication in the Healthcare. 

Sincerely,

Authors.

Reviewer 3 Report (New Reviewer)

Comments and Suggestions for Authors

This is a well described study and a well presented report

I have only one minor remark on language and one more important remark on the presentation.

I believe the word 'fall' should be in singular form when part of a concept, as in 'fall prevention', 'fall rate' and 'fall risk'  (lines 3, 19, 30(Fall rate), 46, 52,  54, 63, 64, 140, 142,142, 146, 161, 163, 259, 293, 318),  and in plural when describing an activity.   Thus: The aim of fall prevention programmes is to lower the risk of falls. 

More important:  Table 1 needs a revision of numbers.

The n of the intervention group is given as 28 in the top line. In second line, 28 are said to be women, which is (87,5%) of 28???   Similar for control group, where 27 is 90% of the 27??? given in the column heading. The same disagreement is seen in all percentages given in the table.

Apparently, the percentages refer to the n at allocation (32,30), whereas the n given in Table refer to the number Post intervention (28,27). 

The whole table should be reconsidered, making sure that the correct numbers of cases  are applied in all calculations.

Also, in line 271, make sure that reference to the number at allocation is the most relevant for the statement.

Comments on the Quality of English Language

English language is fine, except for the detail of the use of singular and plural form of 'fall'

Author Response

Point 1: I believe the word 'fall' should be in singular form when part of a concept, as in 'fall prevention', 'fall rate' and 'fall risk' (lines 3, 19, 30(Fall rate), 46, 52, 54, 63, 64, 140, 142,142, 146, 161, 163, 259, 293, 318), and in plural when describing an activity. Thus: The aim of fall prevention programmes is to lower the risk of falls.

Response: Thank you for your comment. We have revised the plural of the term fall along the manuscript, according to the reviewer's suggestion.

Point 2: Table 1 needs a revision of numbers. The n of the intervention group is given as 28 in the top line. In second line, 28 are said to be women, which is (87,5%) of 28??? Similar for control group, where 27 is 90% of the 27??? given in the column heading. The same disagreement is seen in all percentages given in the table. Apparently, the percentages refer to the n at allocation (32,30), whereas the n given in Table refer to the number Post intervention (28,27). The whole table should be reconsidered, making sure that the correct numbers of cases are applied in all calculations.

Response: Thank you for your comment. The data in the table have been revised and are in accordance with the sample size used in this analysis. We hope that this change will improve the understanding of the article.

Point 3: Also, in line 271, make sure that reference to the number at allocation is the most relevant for the statement.

Response: We have revised this information.

Point 4: Comments on the Quality of English Language: English language is fine, except for the detail of the use of singular and plural form of 'fall'

Response: Thank you for your comment. We hope that the article can contribute positively to the journal. We would like to thank you for allowing us to resubmit our manuscript with all corrections. All the criticisms raised have been carefully addressed, as requested. In the revised manuscript, the changes have been highlighted in red. We believe that the present version of the manuscript has been improved. We are fully available to answer other issues. We hope that now it is suitable for publication in the Healthcare.

This manuscript is a resubmission of an earlier submission. The following is a list of the peer review reports and author responses from that submission.

Round 1

Reviewer 1 Report

Comments and Suggestions for Authors

Effects of a multifactorial program with case management for falls prevention on functional outcomes in community dwelling older people: Randomized Clinical Study

Introduction

Case management is a new approach of multifactorial intervention

Material and methods

It was a randomized trial. Eligible volunteers were Brazilian community-dwelling older adults aged 60 years old and over, non-institutionalized, with a history of falls in the last year. The volunteers were allocating (1:1) in 1 of the 2 groups: Intervention Group (IG) and Control Group (CG). Ethical aspects were observed.

people over 60 are defined as older adults. It is well defined in the article.

Results

They were monitored and evaluated: risk of falls and fractures, physical performance, presence of pain.

Discussion and conclusions

Randomized controlled trial showed that compared to no intervention, 16 weeks of multifactorial falls prevention program based on case management was not effective.

The authors compare the results of the conducted study with other authors and recommend proposals. (for example: home falls prevention Otago exercise program).

I recommend implementing at least 1 or 2 similar studies into the discussion.

I propose to edit the citations according to the latest recommendations. For example, reference:

American College of Sports Medicine. Exercise and physical activity for older adults. Med Sci Sport Exerc. 2009;41:1510–1530

Accept after minor revision (corrections to minor)

Author Response

Response to Reviewer 1 Comments

Point 1: Are the conclusions supported by the results? Can be improved.

Response 1: We would like to thank the Reviewer for allowing us to resubmit our manuscript. All the criticisms raised by the reviwer have been carefully addressed, as requested. In the revised manuscript, the changes are highlighted in red. We believe that the present version of the manuscript has been improved. We hope that now it is suitable for publication in the Healthcare.

Point 2: Discussion and conclusions: Randomized controlled trial showed that compared to no intervention, 16 weeks of multifactorial falls prevention program based on case management was not effective. The authors compare the results of the conducted study with other authors and recommend proposals. (for example: home falls prevention Otago exercise program). I recommend implementing at least 1 or 2 similar studies into the discussion.

Response 2: We appreciated the reviewer's suggestion. Two recently published studies have been added in the discussion section (Line 304-310, Pg 8).

Point 3: I propose to edit the citations according to the latest recommendations. For example, reference:American College of Sports Medicine. Exercise and physical activity for older adults. Med Sci Sport Exerc. 2009;41:1510–1530

Response 3: Thank you for your comment. The references have been reviewed (references section).

Reviewer 2 Report

Comments and Suggestions for Authors

See attached file

Comments on the Quality of English Language

see previous attachment

Author Response

Response to Reviewer 2 Comments

Point 1: It is important to publish manuscripts that do not find a significant diFFerence in intervention and control groups for an intervention study.  The authors need to be more descriptive of the intervention. The case management description is lacking.  In the manuscript the authors note that a third of the intervention group participants did not do the exercises.  Why did this occur?  Did some of the control group participate in a similar exercise program that the intervention group was to participate in?  The manuscript needs to be reviewed for editorial changes throughout.  A few examples are provided in the reviewer comments.

Response 1: We would like to thank the Reviewer for allowing us to resubmit our manuscript. All the criticisms raised by the reviwer have been carefully addressed, as requested. In the revised manuscript, the changes are highlighted in yellow. We believe that the present version of the manuscript has been improved. We hope that now it is suitable for publication in the Healthcare.

Point 2: Line 155: Intervention: It is unclear what was covered in the weekly case management meetings after the first couple of weeks when rapport was established. What was the focus?  Motivational?  The reader is referred to other publications for information about the exercise programs.  Need to provide more about the program in this publication.  Especially since it is reported later that so many of the intervention group participants did not complete the exercise program.

Response 2: We appreciated the reviewer's suggestion. Further details about the intervention have been included in the manuscript, as suggested (line 177-193, Pg 4).

Point 3: Line 171: More information about the case management protocol is described by  [32]. 

Need to add the author name before the reference number.

Response 3:  Thank you for the reviewer's observation. The author's name has been added (line 193, Pg 4).

Point 4: Line 201: At baseline, the mean number of previous falls in the last 12 months, assessed by the  Falls Risk Score, was 2 to 3 falls, which indicates a population at high risk for falls. A mean would not be “2 to 3” falls.  What was the mean – or use a diFFerent measure to indicate that it was 2 to 3.

Response 4: Thank you for the reviewer's comment. The average number of falls in the IG was 2.66was 2.66 in the IG and 3.40 in the CG (line 225-226, Pg 6).

Point 5: Line 209: At baseline, the mean SPPB score was 5 points for both groups, indicating declines in physical performance. The use of the term “declines” is inappropriate.  This is a baseline measure and it is not possible to say that there was a decline at this point in time.

Response 5: We appreciated the reviewer's suggestion. In agreement with the reviewer, we have removed the term (line 233, Pg 6).

Point 6: Line 227 Table 2: Provide a footnote to remind the reader what was the time frame for the short and long term follow up assessments.

Response 6: Thank you for the reviewer's suggestion. A footnote has been added in the table, as suggested (Table 2).

Point 7: Line 252: were not aware of the importance of physical exercise in preventing falls, as 11 of the 32  IG participants (34%) did not agree to participate in the physical exercise program during  the intervention period. With a 1/3 of the participants not doing the exercise program did the authors consider doing statistical analysis excluding this group.  What did this 1/3 group of participants do diAerently that before the case management?  EAicacy of the program needs to be addressed earlier in the manuscript.

Response 7: Thank you for the reviewer's comment. 11 of the 32 IG participants (34%) did not participate fully in the physical exercise program during the intervention period, due to lack of interest and absence of a medical approval, requested in the safety protocol. For these cases, light physical exercises associated with falls prevention and personal interest were recommended. Despite this, it is important to note that the main intervention in this study was the case management program (line 275-281, pg 8).

Point 8: Line 259: Participants completed an average of 23 sessions and 6 reassessments for progression.  Unsure what the 6 reassessments are – as it was mentioned that there was a short and long term assessment.  Not 6.  

Response 8: Thank you for the reviewer's comment. Participants completed an average of 23 sessions and 6 biweekly calls for progression of the physical exercise protocol (line 286-287, pg 8).

Point 9: Line 302 Strengths and Limitations: The authors state: A strength of our study is the remote application of tests and interventions.  How can remote intervention be noted as a strength when 34% of the participants in the IG did not do the exercise program?   The authors do not provide any information on how the remote testing using the various tools compares with testing done “in-person”  If this is considered by the authors as a strength there needs to be more information to support this.

Response 9: Thank you for the reviewer's comment. In fact, we believe that a strength of our study was the remote application of tests and case management interventions. The adaptation to a remote format allowed the project to be carried out during the COVID-19 pandemic, when falls at home increased in Brazil. However, for future studies, it is important to validate the assessments applied remotely in older people with a history of falls (line 335-340, pg 9).

Point 10: Line 313 Conclusions: It is recommended that the authors change the conclusions to address that this study does not support the implementation of the multifactorial falls prevention program as was conducted in this study for the population studied.  Even though evidence supports the use of remote programs to decrease fall risk new strategies to oAer for a frail and high fall risk group need to be studied. The focus on fracture risk and pain not new aspects of an encompassing fall prevention program – so no need to stress these aspects in the conclusions.

Response 10: Thank you for the reviewer's suggestion. The conclusion of the article has been reformulated (line 349-353, pg 9).

Point 11: Edits throughout the manuscript are needed.  Several examples are provided but entire manuscript needs to be reviewed. Line 289: Regarding the Falls Risk Score, this proved to be an important tool that can be used in the educational process of the older people at risk of falls. Rewritten – The Falls Risk Score assessment served as an important tool in the educational process of older people at risk for falls. Line 299: inclusion of assessment and management of pain with a higher sample size “higher” needs to be replaced with “larger”

Response 11: We have edited the manuscript (ln 325-326, pg 9; ln 335, pg 9; along the manuscript).

Reviewer 3 Report

Comments and Suggestions for Authors

INTRODUCTION

-Some statements in the introduction, but also in the discussion section, are not supported by bibliographical references. For example: lines 40-41: “Pain is also a significant risk factor for falls. One in two older people with musculoskeletal pain fall each year.”; lines 52-53: “Multifactorial interventions can prevent and reduce falls risk in community-dwelling older people.”

-Authors should clearly define their research hypothesis.

MATERIALS AND METHODS

-The inclusion/exclusion criteria section should be revised and described more carefully. In detail, “…the willing to participate in the interventions and assessments” is not an inclusion criteria. This is obvious. How could researchers do research without the willingness to participate in research? On the other hand, the authors had previously reported that they had recruited " Eligible volunteers". Moreover, how did the authors evaluate the “ability to walk with or without walking devices”? And how  did the authors establish the “familiarity of the volunteer or his/her family with the use of remote devices (self-reported)”? Furthermore, how were all the disorders reported among the exclusion criteria diagnosed? Were they only self-reported by participants? This section lacks scientific rigor.

-The authors stated that: "The entire research was carried out remotely at the volunteer's home via video calls on Google Meet or telephone". This is the major doubt I have concerning the rigor of the research. In fact, authors below stated that: “The assessment was carried out individually, with the help of a family member or caregiver if necessary, through a video call on google meet”. The beneficial effects of telecoaching are now known and this supports the intervention proposed by the authors. But the fact that the measurements were carried out remotely and with the help of those who do not have expertise in the research field (because they do not do this as a profession) and therefore in the administration of the tests is in my opinion a major criticality of the study. Furthermore, the risk of measurement bias increases.

-Data collection setting needs more details. For example, what time were the measurements taken? This can affect results among participants. What about home setting during the TUG or the SPPB?

DISCUSSION

-Considering that the authors did not find differences between the intervention group and the control group, what are the practical implications of the study? What message do the authors want to convey to the reader? What new knowledge the authors want add to the scientific literature?

Comments on the Quality of English Language

None.

Author Response

Response to Reviewer 3 Comments

Point 1: INTRODUCTION: -Some statements in the introduction, but also in the discussion section, are not supported by bibliographical references. For example: lines 40-41: “Pain is also a significant risk factor for falls. One in two older people with musculoskeletal pain fall each year.”; lines 52-53: “Multifactorial interventions can prevent and reduce falls risk in community-dwelling older people.”

Response 1: We would like to thank the Reviewer for allowing us to resubmit our manuscript. All the criticisms raised by the reviwer have been carefully addressed, as requested. In the revised manuscript, the changes are highlighted in yellow. We believe that the present version of the manuscript has been improved. We hope that now it is suitable for publication in the Healthcare. We have added the references, as suggested (ln 40, pg1; ln 54, pg 2).

Point 2: Authors should clearly define their research hypothesis.

Response 2: We appreciated the reviewer's suggestion. The research hypothesis has been added to the manuscript, as suggested (line 70-74, pg 2).

Point 3: MATERIALS AND METHODS: -The inclusion/exclusion criteria section should be revised and described more carefully. In detail, “…the willing to participate in the interventions and assessments” is not an inclusion criteria. This is obvious. How could researchers do research without the willingness to participate in research? On the other hand, the authors had previously reported that they had recruited " Eligible volunteers". Moreover, how did the authors evaluate the “ability to walk with or without walking devices”? And how  did the authors establish the “familiarity of the volunteer or his/her family with the use of remote devices (self-reported)”? Furthermore, how were all the disorders reported among the exclusion criteria diagnosed? Were they only self-reported by participants? This section lacks scientific rigor.

Response 3: We appreciated the reviewer's observation. In agreement with the reviewer, we have rewritten this section (ln 90-94, pg 2).

 Point 4: The authors stated that: "The entire research was carried out remotely at the volunteer's home via video calls on Google Meet or telephone". This is the major doubt I have concerning the rigor of the research. In fact, authors below stated that: “The assessment was carried out individually, with the help of a family member or caregiver if necessary, through a video call on google meet”. The beneficial effects of telecoaching are now known and this supports the intervention proposed by the authors. But the fact that the measurements were carried out remotely and with the help of those who do not have expertise in the research field (because they do not do this as a profession) and therefore in the administration of the tests is in my opinion a major criticality of the study. Furthermore, the risk of measurement bias increases.

Response 4: We appreciated your feedback. Prior to the assessments, the researchers who administered the tests were trained by a senior researcher with years of experience in administering tests. Also, pilot tests were conducted to minimize any measurement error. Also, support materials were sent to the volunteers to facilitate the use of the remote devices and a previous video call was done to explain better the use. Despite that, we recognized it as a limitation of this study (ln 125-129, pg 3; ln 330-345, pg 9).

Point 5: Data collection setting needs more details. For example, what time were the measurements taken? This can affect results among participants. What about home setting during the TUG or the SPPB?

Response 5: Thank you for your comment. The assessments were conducted remotely during the day and afternoon at a previously scheduled time according to participant availability. Before the assessments, the participants received information about appropriate clothing, the necessary materials, such as a measuring tape, chair and adequate space to perform the tests (ln 125-143, pg 3).

Point 6: DISCUSSION: Considering that the authors did not find differences between the intervention group and the control group, what are the practical implications of the study? What message do the authors want to convey to the reader? What new knowledge the authors want add to the scientific literature?

Response 6: We appreciated the comment. We have rewritten some parts of the discussion and conclusion sections, taking into account the reviewer’s suggestions (ln 260-355, pg 7-9).

Round 2

Reviewer 3 Report

Comments and Suggestions for Authors

The authors did not resolve my doubts. Strong criticisms remain regarding the methodological rigor of the study.

For example, authors stated: "Prior to the assessments, the researchers who administered the tests were trained by a senior researcher with years of experience in administering tests. Also, pilot tests were conducted to minimize any measurement error. Support materials were sent to the volunteers to facilitate the use of the remote devices, materials and space needed and a previous video call was done to explain better the use. The assessment was carried out individually, with the help of a family member or caregiver if necessary, through a video call on Google Meet". It is not clear whether the measurements were carried out by a researcher because, otherwise, it is not explained why a family member or caregiver should have been helped via video call.

Another example, authors stated: "The assessments were conducted remotely during the day or afternoon at a previously scheduled time according to the participant availability". Since the authors measured participants at different times, and not in the same time slot, there may be a time-of-day effect on performance.

For this and the other issues already raised I confirm my previous decision to reject the manuscript.

Comments on the Quality of English Language

None.
